# Acquired *bla*_VIM_ and *bla*_GES_ Carbapenemase-Encoding Genes in *Pseudomonas aeruginosa*: A Seven-Year Survey Highlighting an Increasing Epidemiological Threat

**DOI:** 10.3390/pathogens12101256

**Published:** 2023-10-18

**Authors:** María Guadalupe Martínez-Zavaleta, Diana Fernández-Rodríguez, Melissa Hernández-Durán, Claudia A. Colín-Castro, María de Lourdes García-Hernández, Noé Becerra-Lobato, Rafael Franco-Cendejas, Luis Esaú López-Jácome

**Affiliations:** 1Clinical Microbiology Laboratory, Infectious Diseases Division, Instituto Nacional de Rehabilitación Luis Guillermo Ibarra, Calz, México-Xochimilco No. 289, Col. Arenal de Guadalupe, Mexico City 14389, Mexico; mgmz.am.mm@gmail.com (M.G.M.-Z.); dianafernandezn@hotmail.com (D.F.-R.); melypsp@yahoo.com.mx (M.H.-D.); usedat@gmail.com (C.A.C.-C.); lourdesgh67@gmail.com (M.d.L.G.-H.); noe@ciencias.unam.mx (N.B.-L.); 2Plan de Estudios Combinados en Medicina (PECEM) MD/PhD, Facultad de Medicina, Universidad Nacional Autónoma de México, Circuito Escolar S/N, Ciudad Universitaria, Av. Universidad 3000, Mexico City 04510, Mexico; 3Biomedical Research Subdirection, Research Direction, Instituto Nacional de Rehabilitación Luis Guillermo Ibarra Ibarra, Calz, México-Xochimilco No. 289, Col. Arenal de Guadalupe, Mexico City 14389, Mexico; 4Biology Department, Chemistry Faculty, Universidad Nacional Autónoma de México, Circuito Escolar S/N, Ciudad Universitaria, Av. Universidad 3000, Mexico City 04510, Mexico

**Keywords:** *Pseudomonas aeruginosa*, antimicrobial resistance, carbapenemase, VIM, IMP, GES

## Abstract

(1) Background: *Pseudomonas aeruginosa* is a Gram-negative bacterium with several intrinsic and acquired antimicrobial resistance mechanisms. The spread of carbapenemase-encoding genes, an acquired mechanism, enables carbapenem resistance in clinical settings. Detection of the carbapenemase-producer strains is urgent. Therefore, we aimed to characterize carbapenemase production in the clinical strains of *P. aeruginosa* at a tertiary-care center. (2) Methods: We included clinical strains of *P. aeruginosa* (from August 2011 to December 2018) with resistance towards at least one carbapenem. Strains were isolated in a tertiary-care center in Mexico City. Antimicrobial susceptibility profiles were determined by broth microdilution. Screening for carbapenemase-encoding genes was performed in all strains. Phenotypic assays (CarbaNP and mCIM) were conducted. Additional modifications to mCIM were also tested. (3) Results: One-hundred seventy-one *P. aeruginosa* strains out of 192 included in this study were resistant towards at least one of the carbapenems tested. Forty-seven of these strains harbored a carbapenemase-encoding gene. VIM (59.6%) and GES (23.4%) were the most frequently found carbapenemases in our study, followed by IMP (14.9%). (4) Among the most frequent carbapenemase genes identified, metallo-ß-lactamases were the most prevalent, which impair new treatment options. Searching for carbapenemase genes should be performed in resistant isolates to stop transmission and guide antimicrobial treatment.

## 1. Introduction

*Pseudomonas aeruginosa* belongs to the *aeruginosa* and the Gammaproteobacteria groups, along with *Pseudomonas mendocina*, *Pseudomonas alcaligenes*, *Pseudomonas flavescens*, and other nine species [1]. It is a Gram-negative rod, non-fermenter, indole-negative, oxidase-positive, and mono-flagellated, with its typical “grape-like” or “fresh-tortilla” odor [2]. In addition, it is defined as an opportunistic pathogen and is able to infect plants, animals, insects, and humans [3]. *P. aeruginosa* has been associated with hospital-acquired pneumonia, urinary tract infections, burn wound infections, bloodstream infections, surgical site infections, infectious complications in the immunocompromised, and patients with malignancies [4,5].

These facts could explain the high rates of morbi-mortality associated with the infections of *P. aeruginosa*. According to the Centers for Disease Control and Prevention (CDC), in 2017, an estimated healthcare cost of 767 million dollars, 32,600 in-care cases, and 2700 deaths were attributed to *P. aeruginosa* infections [6]. These costs have certainly increased during COVID-19 pandemic due to the rise in healthcare-associated infections, such as ventilator-associated lower respiratory tract infections, where *P. aeruginosa* highlights as one of the main isolated microorganisms [7].

*P. aeruginosa* features an intrinsic ability to enable infections; moreover, it has inherent resistance mechanisms, and it is able to develop new ones. In 2017, The World Health Organization (WHO) defined *P. aeruginosa* as one of the highest global priority microorganisms; this was due to the resistance mainly associated with the carbapenem family, which are ß-lactams used as antimicrobials of last-resort against *P. aeruginosa* infections [8].

In recent years, carbapenem resistance has been rising around the world [9,10,11,12]. In *P. aeruginosa*, resistance towards carbapenems is explained in two ways. The first one considers structural components and includes loss of porins and over-expression of efflux pumps. Meanwhile, the second one contemplates the production of intrinsic ß-lactamases (e.g., overproduction of AmpC) and acquired enzymes with huge capacity to hydrolyze carbapenems; these latter enzymes are known as carbapenemases [13,14,15].

Carbapenemases hydrolyze carbapenems through their serine residue or the metal cofactor (zinc) within the active catalytic sites, and in addition to being active against carbapenem, they are able to hydrolyze other ß-lactams, with some exceptions [16]. Carbapenemases belonging to classes A, B, and D and have been described to be effective in hydrolyzing a carbapenem that is most commonly used to treat infections by Gram-negatives germs [17]. Overall, class B metallo-ß-lactamases (mainly VIM, IMP, and NDM) have been described as the most frequent carbapenemases produced by *P. aeruginosa* clinical strains. Members of this class demonstrate broad-spectrum hydrolysis of ß-lactams, including carbapenems, as previously mentioned. Additionally, they exhibit an intriguing characteristic that may aid in the phenotypic identification of class members. It should be noted that these enzymes cannot recognize aztreonam as a substrate. In contrast, aztreonam can serve as a reporter for metallo ß-lactamases, but this is not applicable to class A carbapenemases. This discrepancy arises from the ability of class A carbapenemases to hydrolyze aztreonam due to the presence of a serine hydrolytic pocket [16]. However, non-negligible rates of class A carbapenemases (GES and KPC) have been identified worldwide. On the other hand, OXA-type carbapenemases are rare among *P. aeruginosa* clinical isolates [18]. Instead, OXA-type carbapenemases are typically found in the genus Acinetobacter and are referred to as OXA enzymes due to their ability to recognize oxacillin as a substrate. Similar to class A enzymes, these OXA enzymes feature a highly conserved region with a serine residue responsible for the hydrolytic activity [19].

The fast spreading of carbapenemase-encoding genes exhibited among Gram-negative rods, including *Enterobacterales* and non-fermenter bacilli (like *P. aeruginosa*), allows resistance towards carbapenems. This threat poses an urgent need for quick, effective, and updated detection of carbapenemase-producer strains [20] in order to enhance antimicrobial stewardship and guide clinical and epidemiological references. Therefore, we aimed to characterize carbapenemase-production in clinical strains of *P. aeruginosa* at a tertiary-care center in Mexico City.

## 2. Materials and Methods

### 2.1. Clinical Strains, Identification, and Antimicrobial Susceptibility Profiles

We included clinical strains of *P. aeruginosa* from August 2011 to December 2018. They were isolated from patients treated at the National Institute of Rehabilitation Luis Guillermo Ibarra Ibarra, a tertiary-care center specialized in orthopedics, burns, and rehabilitation in Mexico City. Current work was approved by research committee under approval number 88/19 at National Institute of Rehabilitation Luis Guillermo Ibarra and all methods were carried out in accordance with the literature [21]. One randomized strain per patient was included from an institutional database (Infectious diseases laboratory) of positive cultures. Eligible strains were resistant to at least one carbapenem, either meropenem or imipenem. Clinical samples considered for this study were as follows: abscesses, aspirates, biopsies, bloodstream cultures, catheter tips, and urine cultures. These samples were processed according to internal procedures referred in the literature [21].

Strains were first identified with the semi-automated system Vitek 2 (bioMériux; Marcy-l’Étoile, France) and kept at −70 °C until their use, then recovered onto 5% blood sheep agar and checked for purity, then biochemical test such as indole, oxidase, growing at 42 °C and pigment production were performed. Antimicrobial susceptibility profiles were first determined with Vitek 2 and then confirmed by broth microdilution, according to CLSI M07 guidelines [22]. Cutoff values were compared with the CLSI M100 document [23]. Antimicrobials tested at 0.062–64 µg/mL were amikacin (AMK; Sigma Aldrich, Burlington, MA, USA), gentamicin (GEN; Sigma Aldrich, Burlington, MA, USA), aztreonam (ATM; Sigma Aldrich, Burlington, MA, USA), ceftazidime (CAZ; Sigma Aldrich, Burlington, MA, USA), cefepime (FEP; Sigma Aldrich, Burlington, MA, USA), ciprofloxacin (CIP; Sigma Aldrich, Burlington, MA, USA), levofloxacin (LVX; Sigma Aldrich, Burlington, MA, USA), doripenem (DOR; Sigma Aldrich, Burlington, MA, USA), imipenem (IMP; Sigma Aldrich, Burlington, MA, USA), meropenem (MEM; Sigma Aldrich, Burlington, MA, USA), and colistin (COL; Sigma Aldrich, Burlington, MA, USA). Piperacillin/tazobactam (TZP; Sigma Aldrich, Burlington, MA, USA) was tested at 0.25/4–128/4 µg/mL.

### 2.2. Genotypic Detection of Carbapenemases

Carbapenemase-encoding genes were screened in all clinical strains of *P. aeruginosa* included in this study. DNA extraction from a single colony was performed with heat shock at 96 °C during 20 min. Then, the tube was centrifuged at 13,000 rpm for 5 min. Supernatant was collected in a new tube and preserved at −20 °C until its use. Genes, primer sequences, melting temperature (Tm), and amplicon size can be found in Table 1. The 1X ThermoPol (New England Biolabs Inc.; Ipswich, MA, USA) reaction buffer was used for end-point polymerase chain reaction (PCR) according to the manufacturer’s instructions. PCR conditions were as follows: initial denaturation at 95 °C for 1 min, 32 cycles at 95 °C for 32 s, annealing at each pair of primers´ Tm (see Table 1) for 15 s, and extension at 68 °C for 45 s. PCR products were confirmed by gel electrophoresis in a 1% agarose gel, stained with SYBR Green (Thermo Fisher Scientific Inc., Waltham, MA, USA) and run for 1 h at 110 V. Strain P104, a clinical isolate of *P. aeruginosa* from which GES-5 was identified by amplicon sequencing, was used as control for *bla*_GES_ detection. *Enterobacter cloacae* ATCC BAA-2468 and *Klebsiella pneumoniae* ATCC BAA-1905 were used as positive controls for *bla*_NDM_ and *bla*_KPC_, respectively. On the other hand, the pan-susceptible strain, *Escherichia coli* ATCC 25922 was used as negative control. Previously described *P. aeruginosa* clinical strains harboring *bla*_VIM_, *bla*_IMP_, *bla*_OXA-23_, *bla*_OXA-40_, and *bla*_OXA-48_ were also used as positive controls; they were kindly provided by Dr. Garza-Ramos and Dr. Silva-Sánchez from the Instituto Nacional de Salud Pública in Cuernavaca, Morelos, Mexico [24,25,26,27].

### 2.3. Sequencing

PCR products were purified using QIAquick PCR Purification Kit (Qiagen, Hilden, Germany) and quantified in NanoDrop (Thermo Fisher Scientific Inc., Waltham, MA, USA). BigDye Terminator v3.1 cycle sequencing kit (Applied Biosystems, Thermo Fischer Scientific, Waltham, MA, USA) was used and then purified with BigDye Xterminator Purification Kit (Applied Biosystems, Thermo Fischer Scientific, Waltham, MA, USA).

Sequencer used was the 370xl DNA Analyzer (Applied Biosystems, Waltham, MA, USA) and we used POP-7 (Applied Biosystems, Thermo Fisher Scientific, Waltham, MA, USA). Sequences were analyzed using the GenBank database and sequences with each independent oligonucleotide were analyzed.

### 2.4. Phenotypic Detection of Carbapenemases

Carba NP and the modified carbapenem inactivation method (mCIM) were performed to screen for carbapenemases production according to CLSI M100 recommendations for *P. aeruginosa* [23]. *E. cloacae* ATCC BAA-2468, a metallo-ß-lactamase producer, was used as positive control. On the other hand, *E. coli* ATCC 25922, a pan-susceptible reference strain, was used as negative control. All experiments were performed in triplicate.

### 2.5. Statistical Analysis

Eligible strains within the same patient were randomized with http://www.randomization.com accessed on 19 March 2019 [28], and the first strain in each set was selected for experimentation in order to include just one strain per patient. Absolute and relative frequencies were calculated, as well as median and interquartile range (IQR) whenever necessary. Descriptive analysis was performed in STATA 14.0 and graphs were generated in GraphPad Prism 7.0.

## 3. Results

Among the 1551 *P. aeruginosa* strains, isolated between August 2011 to December 2018, we found 411 carbapenem-resistant strains. After the randomization and depuration, we included 192 *P. aeruginosa* carbapenem-resistant strains in this study, each one isolated from a different patient.

*P. aeruginosa* clinical strains included in this study were collected from abscesses (1%), aspirates (11%), biopsies (52%), blood cultures (10%), intravascular catheter tips (5%) and urine (21%).

As seen in Figure 1, the antimicrobial susceptibility profiles showed that at least 89.1% of the strains were resistant to at least one of the carbapenems tested (imipenem, doripenem, meropenem). Almost half of the *P. aeruginosa* strains (44.3%) were susceptible to amikacin; however, resistance to gentamicin was as high as 70.8%. Resistance rates for aztreonam, ceftazidime and cefepime were 54.1%, 73.9% and 76.5%, respectively. Likewise, we found that more than 80% of the strains were resistant to fluoroquinolones, 9.9% were resistant to colistin and 57.3% of the strains showed resistance to piperacillin/tazobactam.

Almost one-fourth (24%, 47/192) of the *P. aeruginosa* strains in this study harbored at least one of the pursued carbapenemase-encoding genes (Figure 2). A class A ß-lactamase-encoding gene, *bla*_GES_, was found in eleven (23.4%) strains, of which *bla*_GES-5_ was the most frequent (4/11). However, the most frequently found genes were the class B, metallo-ß-lactamases: *bla*_VIM_ in twenty-eight (59.6%) strains and *bla*_IMP_ in seven (14.9%) strains. One strain co-harbored *bla*_IMP_ and *bla*_NDM_. The carbapenemase-encoding genes *bla*_KPC_, *bla*_OXA-40_, and *bla*_OXA-48_ were not identified in any of the strains tested. For *bla*_VIM_, *bla*_VIM-2_ was the most frequent, *bla*_VIM-62_ and *bla*_VIM-63_ had only one each; in the case of *bla*_IMP_, this family was over-represented by *bla*_IMP-75_, and finally, the co-producer strain had *bla*_IMP-75_ and *bla*_NDM-1_ (all sequencing results can be found in Appendix A).

The resistance profiles of VIM, GES, IMP, and the co-producer IMP/NDM showed 100% resistance to FEP and CIP, as evident from Figure 3. Additionally, a significant proportion of the GES strains (45.5%) displayed resistance against AMK, whereas 63.6% exhibited resistance towards GEN. Of all the strains carrying VIM, GES, or IMP, the lowest resistance percentages were recorded in COL, ranging from 10% to 18.2%.

The monitoring of the carbapenem-resistant strains and carbapenemase detection throughout the seven years can be seen in Figure 4. The median number of the carbapenem strains was 21.5 (IQR 14.75–33.5) per year. The carbapenem-resistant strains differed between 2011 and 2018; we identified a minimum of ten strains in 2011 and a maximum of forty strains in 2015. The absolute number of strains with metallo-ß-lactamases in 2014 was the highest (seven strains); however, the relative frequency reached 29.1% (eight out of twenty-one) of carbapenem-resistant strains. The lowest occurrence of class B carbapenemases was observed in 2018, with null detection. By contrast, the highest relative frequency was identified in 2011 (50%, 5/10 carbapenem-resistant strains). On the other hand, the highest rate of serine-ß-lactamases was seen in 2013 (13.63%, 3/22) and the lowest was identified in 2011, 2014 and 2016, with null detection.

To confirm the activity of carbapenemases, phenotypic methods were used. Carba NP was the method with the lowest detection rate according to the results of the molecular detection: 16 strains (34%) were positive, 18 (38%) were undetermined, and 13 (28%) were negative. With mCIM, all strains harboring *bla*_GES_ were detected; however, for class B metallo-ß-lactamases, 13.5% (five strains) of VIM producers were not detected, 5.4% (two strains) were undetermined, and 8.1% (three strains) were negative.

## 4. Discussion

Bacterial antimicrobial resistance is a threat for worldwide health-care services. In this regard, WHO released a list of priority pathogens. *P. aeruginosa* was then established as a critical priority microorganism and an urgent call for new strategies was made in order to face infections associated with this bacterium [8]. Additionally, we must consider that *P. aeruginosa* possesses several intrinsic and acquired mechanisms involved in antimicrobial resistance [29]. One of the best studied are carbapenemases, which have caught the attention of specialists and international agencies because of their associated mortality, morbidity, and easy spread within mobile genetic elements like plasmids [18].

In this way, precise and updated antimicrobial resistance surveillance among clinical isolates of *P. aeruginosa* and the involved mechanisms (e.g., carbapenemases) is urgent for the development and the improvement of new and current strategies, respectively. According to data from a national network of epidemiological surveillance (RHOVE, in its Spanish acronym), *P. aeruginosa* was the second most frequent microorganism in the healthcare associated infections, affecting 12.4% of the cases in Mexico [30]. In addition, mortality rates were reported to be as high as 30% in bloodstream infections [31,32]. This study depicts an important asset for the epidemiology of carbapenem-resistant strains of *P. aeruginosa* and the mechanisms of the resistance involved. A previous report about antimicrobial resistance rates by Garza-González et al. [11] showed that carbapenem resistance in Mexico has remained between 27.9–29.4% for *P. aeruginosa* during the last decade. By contrast, analogous regions like Brazil and Iran depicted more alarming figures. Carbapenem resistance rates were identified at up to 60% [33] and mortality rates associated with *P. aeruginosa* carbapenem resistance were as high as 70% [34].

We found less than one-fourth of the clinical strains tested harbored a carbapenemase-encoding gene. Previous reports pointed out that carbapenemase-encoding genes are found in a low proportion of *P. aeruginosa* carbapenem-resistant strains, compared to other Gram-negative bacilli [17]. For example, in a two-year period at three medical German centers, 30% of the *P. aeruginosa* carbapenem-resistant strains had a carbapenemase-encoding gene [35]. These genes’ detection fluctuated in a proportion as low as 4.4% [36] to as high as 30% [35,37]. Throughout the seven-year survey, we identified the highest absolute number of carbapenem-resistant strains in 2016. The next two years (2017 and 2018) were characterized by lower figures (14 and 17 strains, respectively). This trend is consistent with the change in behavior regarding antimicrobial prescription in our hospital: prescriptions required an infectious diseases specialist approval. The new conduct was established as an indirect way of managing antimicrobial resistance rates. Despite this important achievement, Figure 3 exhibits subtle variations in carbapenemase detection, strongly suggesting other antimicrobial resistance mechanisms (e.g., porins and efflux pumps) were involved.

In our work, the most frequently found carbapenemase was VIM, in 59.6% (28) of the strains tested; in addition, VIM-5 was the main enzyme from this group. In second place, we found GES with 23.4% (11), followed by IMP (14.9%, 7). In Mexico, several reports identified carbapenemases belonging to class A serine-ß-lactamases (GES, KPC) and class D ß-lactamases (OXA-23, OXA-24, and OXA-40) in clinical samples. However, similar to our study, the most common carbapenemases belonged to class B metallo-ß-lactamases, with IMP being the most frequent, followed by VIM and NDM [20,38].

Two out of three of the main carbapenemases in our study were class B metallo-ß-lactamases. Likewise, all of them were B1 metallo-ß-lactamases, the subclass containing the most relevant enzymes acquired in clinical settings by *Enterobacterales*, *P. aeruginosa* and other Gram-negative non-fermenters [18]. This finding should be taken into account seriously because newer antimicrobials (e.g., ceftazidime/avibactam or ceftolozane/tazobactam) do not have activity against metallo-ß-lactamases [39]. Moreover, class B metallo-ß-lactamases have been identified in healthcare-associated outbreaks [24,40,41]. Additionally, to date there is not enough information about the performance of ceftazidime/avibactam against GES with carbapenemase activity. These are important concerns and reasons why the efforts regarding characterizing carbapenemase enzymes are important in clinical settings.

We identified GES in a non-negligible proportion of the strains tested, almost a fourth; on the other hand, larger studies have shown smaller figures. Tomomi Hishinuma et al. described 1476 clinical strains of *P. aeruginosa* isolated between 2012 to 2016. In this large study, the authors found the carbapenemase-encoding gene *bla*_GES_ in 9.3% of the strains tested. Moreover, the GES-encoding gene *bla*_GES-5_, known for its carbapenemase activity, was found in 75.9% (104/172) of the strains with a GES carbapenemase [42].

These latter facts also support the great concern and interest to develop and apply methodologies, phenotypic or genotypic, able to detect carbapenem-producer strains. In this study, all clinical strains of *P. aeruginosa* that harbored a carbapenemase-encoding gene were tested for carbapenemase production by conventional phenotypic tests (CarbaNP and mCIM) [23]. Our work exhibited that these enzymes were not always shown in these phenotypic assays. We were not able to verify carbapenemases production in five strains, two were considered undetermined, and three were negative.

There are two main phenotypic strategies recommended in previous studies and by international agencies, such as CLSI and European Committee on Antimicrobial Susceptibility Testing (EUCAST): CarbaNP and mCIM. According to CLSI, the sensitivity and specificity of CarbaNP are superior to 90% for *P. aeruginosa*, making it the weakest recommended methodology for carbapenemase production detection [23]. Meanwhile, mCIM shows sensitivity and specificity greater than 97% and 100%, respectively. Despite these figures, there are some factors to be considered that may modify the results, e.g., capsule hyperproduction, mucosity, and low hydrolytic activity [17,43].

Our study has some limitations. First, we did not apply any molecular tool to identify epidemic high clones of *P. aeruginosa* within our clinical strains; however, we included a unique randomized strain per patient to reduce the bias. Moreover, carbapenemase production detection was guided with the molecular results. Therefore, we included primers targeting the most frequent carbapenemase genes described in the literature. Nevertheless, a low number of strains may have harbored unusual carbapenemase-encoding genes. We did not perform broth microdilution susceptibility test for the newer antimicrobial agents.

This study should incentivize epidemiological surveillance of carbapenem resistance in *P. aeruginosa* clinical strains, as well as the unmasking of underlying antimicrobial resistance mechanisms, especially those involved in antimicrobial prescription. Further studies should also focus on the methodological caveats regarding the detection of carbapenemase production.

## 5. Conclusions

The most frequent carbapenemases in our population were VIM and GES. GES represents a niche of opportunities because there are no commercial strategies, neither phenotypic nor genotypic, to detect them. At least in our population, they represent an important frequency that should not be underestimated and should continue to be monitored, as well as to urge to look for it in the rest of the laboratories due to the possible implications that they could have.

## Figures and Tables

**Figure 1 pathogens-12-01256-f001:**
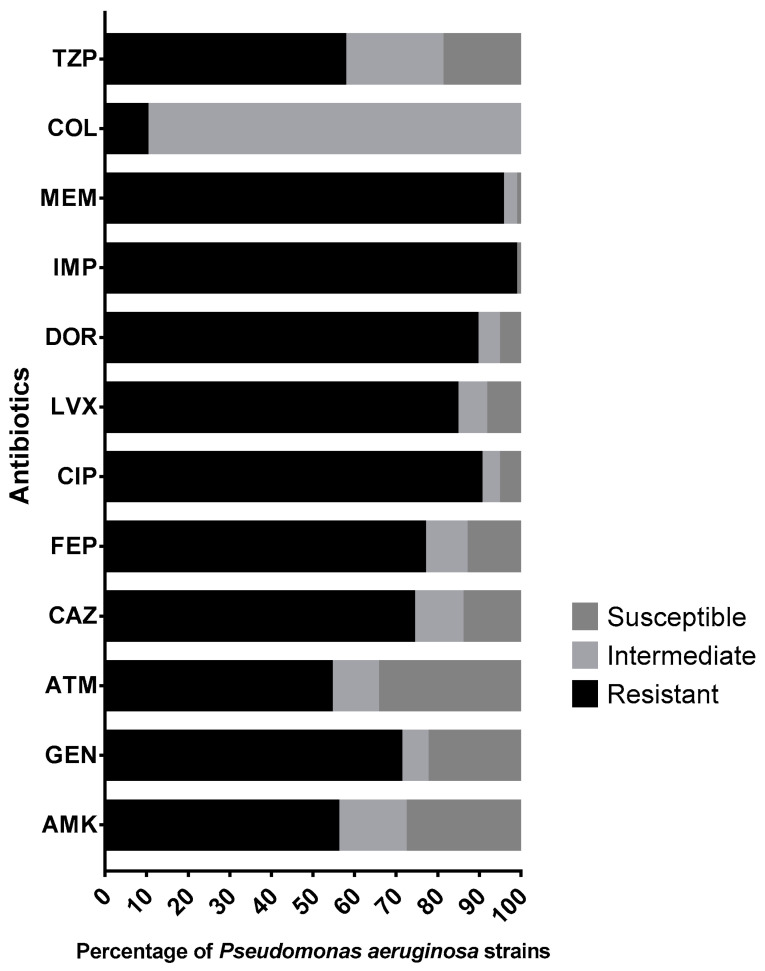
Antimicrobial susceptibility profiles of 192 *Pseudomonas aeruginosa* clinical strains. Abbreviations: TZP, piperacillin/tazobactam; COL, colistin; MEM, meropenem; IPM, imipenem; DOR, doripenem; LVX, levofloxacin; CIP, ciprofloxacin; FEP, cefepime; CAZ, ceftazidime; ATM, aztreonam; GEN, gentamicin; AMK, amikacin.

**Figure 2 pathogens-12-01256-f002:**
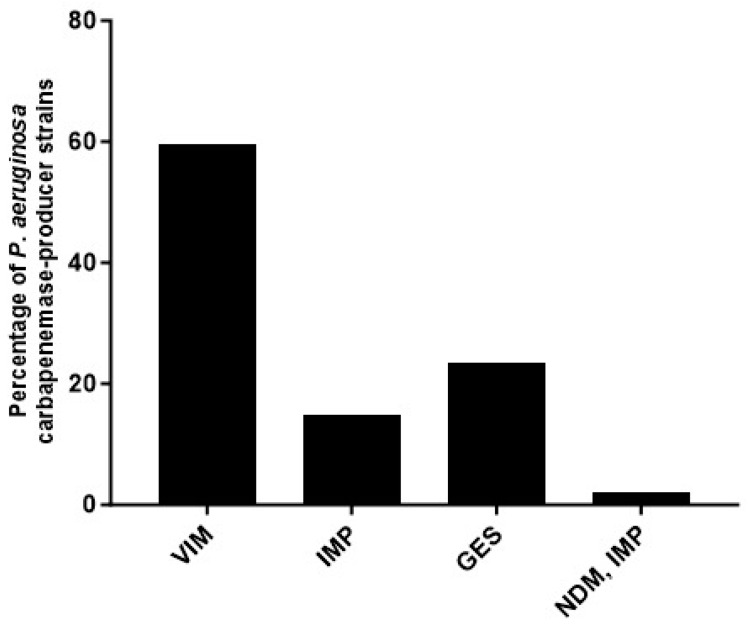
Percentage of *Pseudomonas aeruginosa* clinical strains producing or co-producing carbapenemases.

**Figure 3 pathogens-12-01256-f003:**
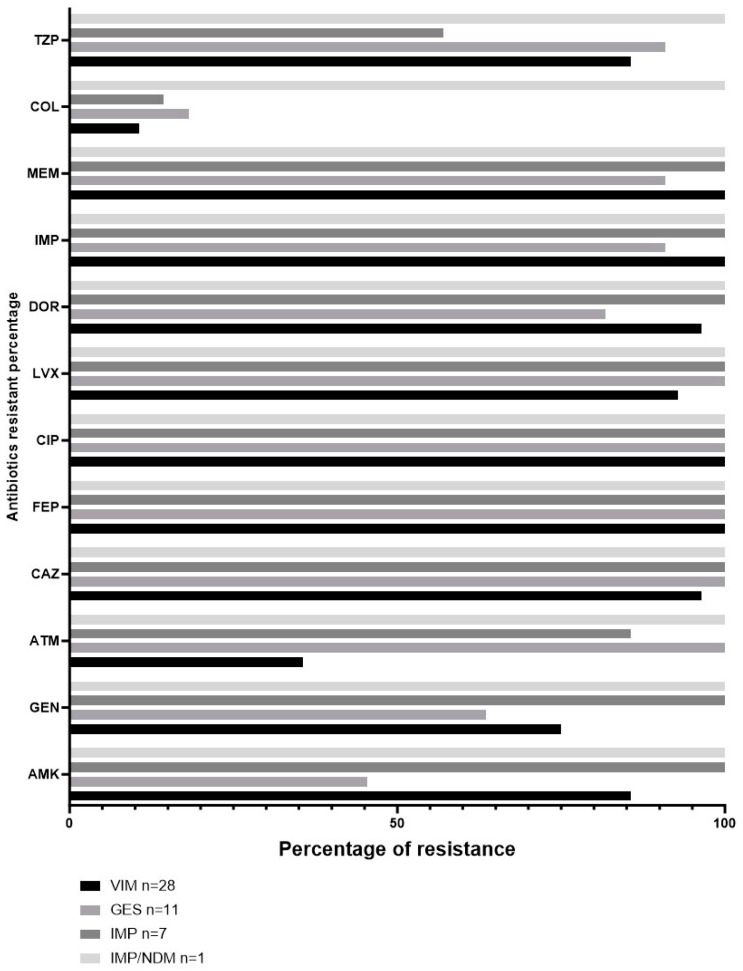
Percentage of antibiotic resistance according to carbapenemase detected in clinical strains of *Pseudomonas aeruginosa*.

**Figure 4 pathogens-12-01256-f004:**
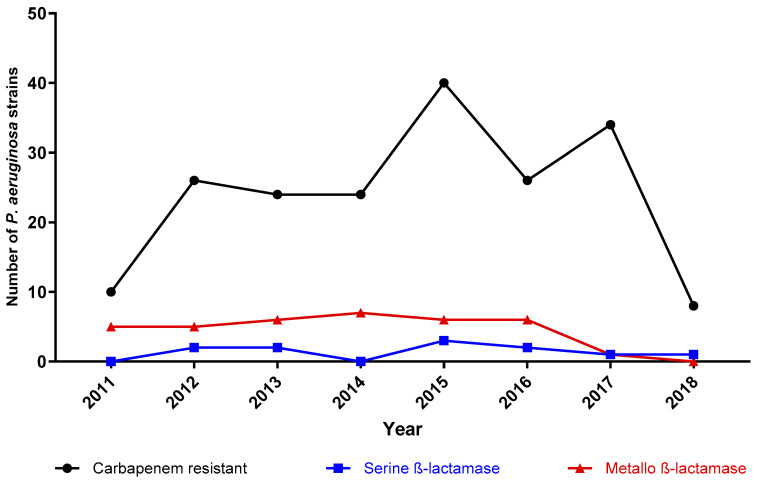
Carbapenem-resistant and carbapenemase-harboring strains throughout the seven-year survey. The black line with triangles represents the absolute number of carbapenem-resistant strains. Strains with metallo-ß-lactamases and serine-ß-lactamases are pictured by the red line with circles and the blue line with squares, respectively. Each figure (triangle, circle, and square) represents a specific year according to the *x*-axis.

**Table 1 pathogens-12-01256-t001:** Primer sequences of carbapenemase-encoding genes used in this study.

Gene	Primer Sequences	Tm (°C)	Amplicon Size (bp)
*bla* _NDM_	F: 5′-ATGGAATTGCCGAATATT-3′	56	~600
	R: 5′-TCAGYGCAGCTTGTCGGC-3′		
*bla* _IMP_	F: 5′-GTTTATGTTCATACTTCGTTTG-3′	52	~400
	R: 5′-CAACCAGTTTTGCHTTAC-3′		
*bla* _VIM_	F: 5′-AGATTGVCGATGGTGTTTGGT-3′	56	~400
	R: 5′-GAGCAAGTCTAGACCGCCC-3′		
*bla* _KPC_	F: 5′-ATGTCACTGTATCGCCGTCT-3′	56	798
	R: 5′-TTACTGCCCGTTGACGC-3′		
*bla* _GES_	F: 5′-TCATTCACGCHCTATTVCTGGCA-3′	58	857
	R: 5′-CTATTTGTCCGTGCTCAGG-3′		
*bla* _OXA-23_	F: 5′-TCTGGTTGTACGGTTCA-3′	56	~300
	R: 5′-TCATTACGTATAGATGCC-3′		
*bla* _OXA-40_	F: 5′-TGAAGCTCAAACACAGGG-3′	56	~400
	R: 5′-AACACCCATTCCCCATCC-3′		
*bla* _OXA-48_	F: 5′-GAATGCCTGCGGTAGCAA-3′	56	438
	R: 5′-AAACCATCCGATGTGGGCAT-3′		

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
