# Peer review of "Acquired blaVIM and blaGES Carbapenemase-Encoding Genes in Pseudomonas aeruginosa: A Seven-Year Survey Highlighting an Increasing Epidemiological Threat"

_pathogens, 2023, doi:10.3390/pathogens12101256_

Round 1

Reviewer 1 Report

The manuscript “Acquired blaVIM and blaGES Carbapenemase Encoding Genes in Pseudomonas aeruginosa: a Seven-Year Survey High-3 lighting an Increasing Epidemiological Threat” characterized the P. aeruginosa clinical strains collected from the patients of a tertiary-care center specialized in orthopedics, burns, and rehabilitation in Mexico City in 2011-2018. The study includes obtaining of the antimicrobial resistance phenotype and detection of carbapenemase genes. Unfortunately, genetic clones of the strains were not identified.

Minor correction should be done in the text.

Comments: 

Question: Why blaIMP and blaMDM genes were missed in the Title?

Lines 35, 36. VIM (n=…), GES (n=…). IMP (n=…), NDM (n=…) should be added.

Line 79. “other” should be deleted.

Line 115. polymerase chain reaction (PCR) reaction, according

Lines 122-125. Correct gene names, please (blaNDM  etc.), in all parts of the manuscript.

Lines 130-135. City of the producers should be presented.

Figure 2. Correct “Porcentage”, please. “Pseudomonas aeruginosa” must be in italic.

Table 1. Correct “blaKCP” on “blaKPC”, please.

Lines 248-260б and in all parts of Manuscript. Numerical values must be represented with the same number of decimal places.

Lines 257-258. It should be better as: “… in Enterobacterales, P. aeruginosa, and other nonfermenting Gram-negative bacteria”

Lines 287-288. Authors note that they “did not identify epidemic-high clones of P. aeruginosa within our clinical strains”. Actually, they did not identify the clinal identity (sequence type or clonal complex) of the strains in this study. 

Author Response

Thank you very much for your effort and your time.

Reviewer 2 Report

The current study determines the incidence of different B-lactamases in P. aeruginosa. There are some points to be addressed.

- Add a figure representing the B-lactamase-producing pseudomonas among the whole collected samples

- Correlate between the phenotypic resistance patterns and the incidence of different B-lactamase encoding genes

- Detail the different types of B-lactamases in the introduction

- Add conclusion

- Check all bacteria names are in italic

- Remove word "title" from the study title

ok

Round 2

Reviewer 2 Report

The authors addressed the raised points

Ok